# Information-Theoretic Distillation for Reference-less Summarization

**Jaehun Jung**[†]    **Ximing Lu**[†]    **Liwei Jiang**[†]    **Faeze Brahman**[†‡]    **Peter West**[†]
**Pang Wei Koh**[†]    **Yejin Choi**[†‡]
[†]Paul G. Allen School of Computer Science & Engineering, University of Washington
[‡]Allen Institute for Artificial Intelligence
hoony123@cs.washington.edu

## Abstract

The current winning recipe for automatic summarization is using proprietary large-scale language models (LLMs) such as ChatGPT as is, or imitation learning from them as teacher models. While increasingly ubiquitous dependence on such large-scale language models is convenient, there remains an important question of whether small-scale models could have achieved competitive results, if we were to seek an alternative learning method—that allows for a more cost-efficient, controllable, yet powerful summarizer. We present INFOSUMM, a novel framework to distill a powerful summarizer based on the information-theoretic objective for summarization, without relying on either the LLM's capability or human-written references. To achieve this, we first propose a novel formulation of the desiderata of summarization (saliency, faithfulness and brevity) through the lens of mutual information between the original document and the summary. Based on this formulation, we start off from Pythia-2.8B as the teacher model, which is not yet capable of summarization, then self-train the model to optimize for the information-centric measures of ideal summaries. Distilling from the improved teacher, we arrive at a compact but powerful summarizer with only 568M parameters that performs competitively against ChatGPT, without ever relying on ChatGPT's capabilities. Extensive analysis demonstrates that our approach outperforms in-domain supervised models in human evaluation, let alone state-of-the-art unsupervised methods, and wins over ChatGPT in controllable summarization.

## 1 Introduction

The winning recipe for summarization today is to prompt a gigantic, proprietary LLM such as ChatGPT, either as a summarizer itself or as a teacher model for imitation learning (Goyal et al., 2023). In order to reduce the inference cost, one maybe particularly tempted to distill a compact summarizer from the LLM: by collecting some documents, instructing the LLM to summarize them, and supervising a small model to simply imitate the generations (Xu et al., 2023; Sclar et al., 2022). Despite its intuitive appeal, this process does not involve how we explicitly define a good summary—the feasibility of data generation is fundamentally dependent on the LLM's capability to follow the instruction. With no quantifiable objective for summarization, our best option is to use the largest and strongest LLM as the teacher, and enumerate as much imitation data as possible from it (Li et al., 2023; Mukherjee et al., 2023). Despite this increasing dependence on large LMs, it is still unclear whether the distilled summarizer will fully generalize across diverse use cases (Gudibande et al., 2023), whether it be zero-shot adaptation to unseen domains or generating controllable summaries.

In this work, we shift our attention from using a larger and stronger teacher model, and show that even the small, off-the-shelf LMs can teach themselves to excel at summarization, provided we define an information-theoretic objective for summarization. Concretely, we propose that the three evaluative dimensions of summarization—saliency, faithfulness and brevity—can be incorporated into a unified search objective, where we look for a summary

**1. Information-maximizing objective for Summarization**

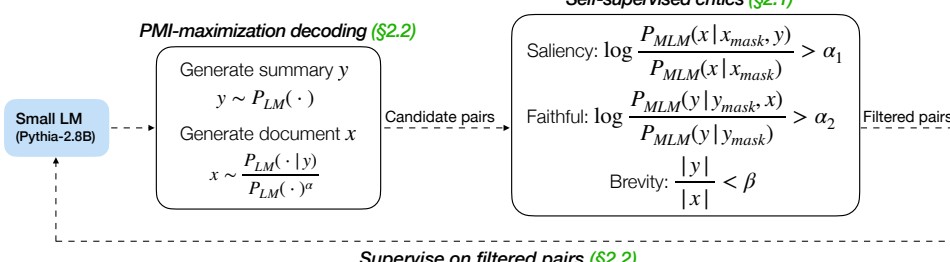

**2. Self-Improve teacher through Expert Iteration**

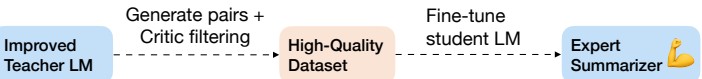

**3. Distill a student summarizer**

Figure 1: **Overview of INFOSUMM.** We formulate summarization as (1) information maximization objective under a length constraint, which allows us to (2) self-train an expert teacher from only a small, off-the-shelf LM and self-supervised critics. Finally, (3) distilling from the improved teacher, we obtain a compact yet powerful summarizer **without relying on an LLM already competent at summarization or human-annotated references**.

$y$ that maximizes its point-wise mutual information (PMI) with the document $x$ under a length constraint. By self-improving the teacher through expert iteration (Anthony et al., 2017) to align with our objective, we yield a high-quality summarization dataset only from a small teacher LM not tuned for summarization. This method, **INFOSUMM** (Figure 1), decouples *what we expect to generate* (*i.e.,* the explicit search objective for summarization) from *how we generate them* (*i.e.,* data-generating LM), allowing us to distill a powerful summarization model without human-written references or an LLM already competent at summarization. Compared to a prior work that distills from a small, off-the-shelf LM (Jung et al., 2023), **INFOSUMM** targets substantially longer, document-level summarization, and operates entirely without human-supervised critics.

Applying our method, we train a 568M summarizer with the dataset generated from Pythia-2.8B (Biderman et al., 2023), an off-the-shelf autoregressive LM that itself cannot reliably summarize a given document. We test our model on diverse tasks spanning news summarization, zero-shot generalization to unseen domains, and controllable summarization. Our model, despite its small scale, exhibits surprisingly strong performance compared to the state-of-the-art: it significantly outperforms all unsupervised methods in reference-based evaluation, improving more than 2 R-1 points across benchmarks. In GPT-4 and human evaluation, our system is even preferred to the in-domain reference-supervised models, and outperforms 175B ChatGPT with a simple re-ranking approach. Notably, our model, as a compact, expert model for summarization, exhibits significantly better controllability than prompting ChatGPT (*e.g.,* to generate a long, highly specific summary of the given document), establishing a promising alternative to imitating human references or LLM-generated summaries.

## 2 INFOSUMM: Information-Theoretic Distillation for Summarization

### 2.1 Summarization as Information Maximization

Intuitively, a good summary $y$ should be a brief representation of the original document $x$ *(brevity)*, that focuses on the key information of $x$ *(saliency)*, without hallucinating unsup-

ported content *(faithfulness)* (Fabbri et al., 2021b). In this section, we first quantify these desiderata of summarization in an information-theoretic perspective, then discuss how they can be unified as maximizing the mutual information between the document $x$ and summary $y$ subject to a length constraint on $y$.

**Saliency** A good summary $y$ should well represent the salient information of the document; information-wise, it should effectively reduce the uncertainty of document $x$ without directly observing it. To empirically measure saliency, we employ a masked language model (MLM)— by masking the tokens in the document $x$ to produce $x_{mask}$, then measuring how well an MLM can recover $x$ from $x_{mask}$ when given the summary $y$. Leveraging this idea, we introduce a saliency critic $f_S$:

$$f_S(x,y) = \mathbb{1}\left\{\log \frac{p_{MLM}(x|x_{mask}, y)}{p_{MLM}(x|x_{mask})} > \tau_S \frac{|y|}{|x|}\right\} \tag{1}$$

Concretely, we normalize the score with the likelihood of reconstructing $x$ from $x_{mask}$ without $y$, as some masks maybe easily inferred without observing $y$.[1] The critic operates by filtering out $(x,y)$ pairs with the normalized score lower than a threshold, where $\tau_S$ is a hyperparameter. The compression ratio $|y|/|x|$ in the threshold reflects the trade-off between summary length and saliency—*i.e.,* a longer summary should better preserve the information of the document. Notably, the proposed critic requires only a self-supervised MLM as a proxy model, as opposed to human-supervised critics in Jung et al. (2023)[2].

**Faithfulness** Neural summarizers often suffer from hallucination, *i.e.,* adding information to the summary that was not present in the original document (Chen et al., 2022; Laban et al., 2022). Under our formulation, faithfulness can be measured in a reverse direction of saliency, by recovering the summary $y$ from $y_{mask}$ given the document $x$:

$$f_F(x,y) = \mathbb{1}\left\{\log \frac{p_{MLM}(y|y_{mask}, x)}{p_{MLM}(y|y_{mask})} > \tau_F\right\} \tag{2}$$

Intuitively, masks on $y$ would be easier to infer given $x$, if $y$ did not add additional information not in $x$. Unlike the saliency critic, we do not include the compression ratio in the filtering threshold, as we expect a good summary to be always faithful to the document regardless of its length.

**Brevity** Finally, a summary $y$ should be a brief representation of the input $x$, evaluated by the compression ratio between the summary and the document:

$$f_B(x,y) = \mathbb{1}\left\{\frac{|y|}{|x|} < \tau_B\right\} \tag{3}$$

**Information-Maximizing Objective** Essentially, the saliency and faithfulness critics can both be considered as filtering based on the PMI between $x$ and $y$, where the two critics differ in how they approximate the mutual information. Specifically, in case of the saliency critic,

$$\log \frac{p_{MLM}(x|x_{mask}, y)}{p_{MLM}(x|x_{mask})} \approx \log \frac{p(x|y)}{p(x)} = \log \frac{p(x,y)}{p(x)p(y)} = \text{PMI}(x; y) \tag{4}$$

The same derivation applies to faithfulness critic. Therefore, incorporating all 3 dimensions above, our objective for distilling text-summary pairs can be crisply described as

> *Searching for a pair* $(\mathbf{x}, \mathbf{y})$ *of fluent text that maximizes its mutual information* *PMI$(\mathbf{x}; \mathbf{y})$, subject to* $\frac{|y|}{|x|} < \tau_B$.

Broadly seen, the mutual information between the input data and its compression have been utilized in prior works, primarily as a metric for unsupervised text evaluation (Kim

---

[1]While we do not require a specific masking strategy to produce $x_{mask}$, we find that masking salient keywords identified by TF-IDF (Laban et al., 2020) allows efficient approximation in practice.

[2]Specifically, Jung et al. (2023) employs a supervised NLI model as a critic for sentence summarization, which does not generalize well to document-level inputs.

et al., 2022; Vasilyev et al., 2020) and feature extraction (Padmakumar & He, 2021; Covert et al., 2023). Compared to these approaches, we formulate PMI maximization as a unified objective for abstractive summarization, which can be optimized even with an off-the-shelf LM by rejection sampling with the self-supervised critics defined above.

## 2.2 From Off-the-shelf LM to Expert Summarization Model

Our goal in INFOSUMM is to start from a small, off-the-shelf teacher LM $M_{init}$, then generate a large-scale summarization dataset $D_{summ}$, which we use to distill an expert summarizer $M_{summ}$. Our key idea is to self-train the teacher $M_{init}$ through expert iteration (Anthony et al., 2017) to align with our information-maximizing objective, yielding an improved data generator $M_T$ for summarization prior to distilling a student.

**Generating Initial Dataset** We start by generating an initial dataset $D_{init}$ from the off-the-shelf teacher $M_{init}$, by over-generating candidate document-summary pairs with the teacher and subsequently filtering them using the critics defined in §2.1.

To generate the candidate pairs, we take a simple auto-regressive approach—we first sample text from $M_{init}$, then just take 1-5 leading sentences as a summary of the remaining content, *i.e.*, $y \sim p_{init}(\cdot|P)$, $x \sim p_{init}(\cdot|P, y)$. Here, $P$ is a simple prefix for better generation quality (*e.g., New York, (CNN)* – to promote news-style generation). We find this approach particularly effective for two reasons. First, it is an easy way to condition the generation of $x$ on $y$, without fine-tuning the autoregressive teacher $M_{init}$. Next, the leading sentences often contain the most salient information of the document, hence have been used as a useful proxy of summary in previous works (*e.g.,* Zhu et al. (2019)).

A limitation of the autoregressive approach, however, is that it generates a long document conditioned on a handful of sentences in the beginning – as the generation gets longer, it easily loses coherence from the original context. To mitigate this issue, we devise a decoding algorithm inspired by *Product of Experts* (Hinton, 2002; Liu et al., 2021) on top of $M_{init}$:

$$y \sim p_{init}(\cdot|P), \quad x \sim p_{init}(\cdot|P, y)p_{init}(\cdot)^{-\alpha}, \ \alpha > 0 \tag{5}$$

By penalizing the unconditional likelihood of the document, we encourage the teacher to attend more to the leading sentences $y$ while generating $x$. Note that if we set $\alpha = 1$, $x \sim \frac{p_{LM}(\cdot|P,y)}{p_{LM}(\cdot)}$, therefore $x$ is generated to maximize its PMI with the summary $y$. Using the decoding algorithm, we distill the initial dataset $D_{init}$ by over-generating candidate set $C_{init}$ of document-summary pairs, then filtering it using the critics defined in §2.1:

$$C_{init} = \{(x_1, y_1), \cdots | y_i \sim p_{init}(\cdot|P), \ x_i \sim p_{init}(\cdot|P, y_i)p_{init}(\cdot)^{-\alpha}\} \tag{6}$$

$$D_{init} = \{(x, y)|(x, y) \in C_{init}; \ f_S(x, y) \land f_F(x, y) \land f_B(x, y) = 1\} \tag{7}$$

**Distilling Expert Summarizer** While the critic models and decoding algorithm effectively implement our search objective, the sampling efficiency of the generated pairs is of central concern when distilling a large-scale dataset. That is, most candidate pairs in $C_{init}$ are unlikely to pass the critic filtering stage, as our initial teacher model is assumed to be not aligned for summarization.

To resolve the bottleneck of low sampling efficiency, we perform a loop of *expert iteration* (Anthony et al., 2017; Silver et al., 2017) on the teacher $M_{init}$, where the off-the-shelf LM is supervised on its own generated, high-quality pairs. Concretely, instead of distilling a student summarizer $M_{summ}$ with $D_{init}$, we self-train the teacher $M_{init}$ to maximize $E_{(x,y) \sim D_{init}}[\log p_{M_{init}}(y, x|P)]$, yielding an improved teacher $M_T$. By training exclusively on high-quality pairs identified by the critics, the teacher model is gradually aligned with our search objective; as we show in the later section, even a single step of expert iteration dramatically boosts the sampling efficiency of the teacher model. Compared to previous works on expert iteration, we yield a specialized data generator entirely from pre-trained LMs, without resorting to ground-truth references (Zelikman et al., 2022) or a human-supervised verifier (Lightman et al., 2023).

Finally, we distill an expert summarizer $M_{summ}$ from the improved teacher $M_T$. First, a large-scale dataset $D_{summ}$ is distilled from the improved teacher $M_T$, following the same

| Dataset | CNN/DM | | | | | XSUM | | | | |
|---|---|---|---|---|---|---|---|---|---|---|
| Model | R-1 | R-2 | R-L | BERTScore | G-Eval | R-1 | R-2 | R-L | BERTScore | G-Eval |
| **In-domain Supervision** | | | | | | | | | | |
| PEGASUS$_{SFT}$ | 44.2 | 21.5 | 41.1 | 88.4 | 4.14 | 47.2 | 24.6 | 39.3 | 91.4 | 4.05 |
| **Unsupervised Methods** | | | | | | | | | | |
| TL;DR | 20.1 | 5.5 | 19.6 | 74.9 | 3.12 | 10.8 | 1.3 | 8.3 | 77.8 | 2.32 |
| ChatGPT | 35.0 | 13.6 | 28.2 | 86.4 | **4.47** | 30.6 | 9.8 | 22.5 | 84.7 | **4.45** |
| SEQ$^3$ | 23.2 | 7.1 | 22.2 | 81.6 | 3.52 | 12.3 | 1.5 | 10.7 | 80.4 | 2.41 |
| Summary Loop | 37.7 | 14.8 | 34.7 | 83.2 | 3.81 | 11.7 | 1.5 | 9.0 | 79.2 | 2.75 |
| TED | 38.7 | 16.8 | 35.4 | - | - | - | - | - | - | - |
| WikiTransfer | 40.1 | 17.7 | 36.7 | - | - | 31.9 | 10.4 | 23.8 | - | - |
| INFOSUMM-0.5B | **42.0** | **19.4** | **38.4** | **88.1** | 4.38 | 33.4 | 14.0 | 28.2 | 85.5 | 4.21 |

Table 1: Quantitative results on news summarization. **INFOSUMM-0.5B, our 568M model distilled from a 2.8B teacher, achieves comparable zero-shot performance to prompting ChatGPT.** For G-Eval, we use GPT-4 evaluation based on a 1-5 Likert scale, averaged across the 4 evaluation criteria *(coherence, consistency, fluency, and relevance)* proposed in the original paper (Liu et al., 2023b). Note that G-Eval is known to have preference bias towards summaries from ChatGPT.

process with $D_{init}$. Then, we fine-tune a student LM into an expert summarizer $M_{summ}$ by maximizing $E_{(x,y) \sim D_{summ}}[\log p_{M_{summ}}(y|x)]$, *i.e.,* the conditional log-likelihood of $y$ given $x$. As a byproduct of this process, we obtain a large-scale, high-quality summarization dataset $D_{summ}$ that can be interpreted and reused, *e.g.,* to train a summarization model without re-iterating the overall distillation process.

**Endowing Controllability**   Controllable summarization has recently emerged as an important research direction (Fan et al., 2018; He et al., 2022), allowing users to customize diverse aspects of the generated summary (*e.g.,* length and specificity). Under INFOSUMM, a controllable summarizer can be trained simply by post-hoc annotating the generated data with the control attributes (for more details, see Appendix B). Moreover, as our framework operates with a small LM as a data generator, we can down-sample over-generated pairs to increase the diversity of control attributes. After annotating control attributes, a student can be trained to be controllable by prepending the control code (Keskar et al., 2019) as an instruction to the input document.

## 3   Experimental Results

### 3.1   Implementation Details

We start from Pythia-2.8B, an off-the-shelf decoder-only LM as the teacher model. Using T5-large (Raffel et al., 2023) as the MLM in the critics, we generate an initial dataset $D_{init}$ of 140K news style text-summary pairs. After self-training the teacher model, we generate a large scale dataset $D_{summ}$ with 4.5M samples, among which 1M pairs are additionally annotated for controllability. In our experiments, we focus on 4 dimensions of control attributes – length, extractiveness, specificity, and keywords – proposed in Zhang et al. (2023b). We also consider a composition of these control attributes, hence allowing the distilled model to follow fine-grained instructions (*e.g., to generate a highly specific, medium length summary focusing on a given keyword*). Using this dataset, we train PEGASUS (Zhang et al., 2020a) with 568M parameters into an expert summarization model. We refer to this model as INFOSUMM-0.5B. Further implementation details are in Appendix A.

### 3.2   Zero-shot News Summarization

**Evaluation Setup**   We first test INFOSUMM-0.5B for zero-shot summarization on widely used benchmarks, XSUM (Narayan et al., 2018) and CNN/DM (Nallapati et al., 2016). For baselines, we consider state-of-the-art unsupervised summarizers (*i.e.,* trained without human references)—TL;DR prompting (Radford et al., 2019) on Pythia-2.8B, SEQ$^3$ (Baziotis et al., 2019), Summary Loop (Laban et al., 2020), TED (Yang et al., 2020) and WikiTransfer

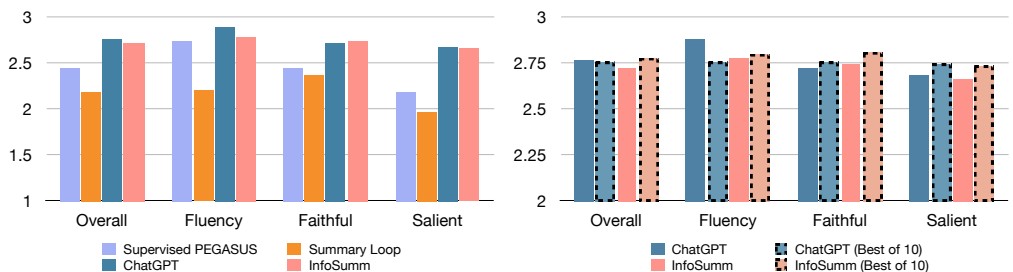

Figure 2: Human evaluation results. **INFOSUMM-0.5B is consistently scored higher than in-domain supervised PEGASUS, and outperforms ChatGPT with a simple re-ranking approach** (*best-of-10*). Left: We compare INFOSUMM-0.5B against baselines across 4 dimensions of summary quality. Right: We test *best-of-10* approach on top of ChatGPT and INFOSUMM-0.5B, by sampling 10 summaries per document and ranking them using the critic models of INFOSUMM.

| Dataset | WikiHow | | Reddit TIFU | |
|---|---|---|---|---|
| Model | *R-L* | *G-E* | *R-L* | *G-E* |
| Summary Loop | 20.0 | 3.59 | 12.4 | 2.55 |
| PEGASUS$_{CNN/DM}$ | 19.5 | 3.67 | 17.7 | 4.19 |
| ChatGPT | 21.1 | **4.45** | **19.8** | **4.38** |
| INFOSUMM-0.5B | **22.9** | 4.35 | 17.9 | 4.26 |

Table 2: **INFOSUMM better generalizes to unseen domains than human-supervised PEGASUS.** We report ROUGE-L (R-L) and G-Eval (G-E) on WikiHow and Reddit domains.

| Dataset | WikiHow | | Reddit TIFU | |
|---|---|---|---|---|
| Model | *R-L* | *B-S* | *R-L* | *B-S* |
| PEGASUS$_{SFT}$ | 34.8 | 88.8 | 21.6 | 87.6 |
| PEGASUS$_{CNN/DM-SFT}$ | 33.8 | 87.4 | 21.1 | 87.2 |
| INFOSUMM-0.5B $_{SFT}$ | **35.2** | **90.0** | **23.0** | **87.7** |

Table 3: **INFOSUMM is effective for transfer learning.** After fine-tuning, our model better matches the reference of each benchmark than PEGASUS, measured by ROUGE-L (R-L) and BERTScore (B-S).

Fabbri et al. (2021a), as well as zero-shot prompted ChatGPT (*gpt-3.5-turbo*). We also consider an in-domain supervised baseline PEGASUS$_{SFT}$, fine-tuned on the respective train sets of the benchmarks. For metrics, we report ROUGE, BERTScore (Zhang et al., 2020b) and average G-EVAL (Liu et al., 2023b), a reference-less metric based on GPT-4 known to better correlate with human judgements of summary quality.

**Results** We present our quantitative results in Table 1. INFOSUMM-0.5B significantly outperforms summarization-specific unsupervised baselines – including Summary Loop trained with in-domain articles of CNN/DM, and WikiTransfer that leverages the stylistic bias of each benchmark (hence not considered to be strictly zero-shot). Overall, our model marks similar performance across metrics with ChatGPT, an order of magnitude larger, human-aligned LLM. In GPT-4 evaluation on XSUM, it even outperforms PEGASUS$_{SFT}$, the same base model PEGASUS supervised on the in-domain references.

**Human Evaluation** To better compare the quality of model-generated summaries, we conduct human evaluation. We generate summaries for 200 CNN/DM articles with each system, then ask 6 annotators to score their fluency, faithfulness and saliency following Stiennon et al. (2022). To adjust the confounding effect of summary length, we sample only those articles for which all systems output summaries with the same number of sentences. To minimize subjectivity, we use strict 3-level Likert scale, leading to high inter-annotator agreement (Krippendorff's alpha=0.65; substantial agreement).

The left part of Figure 2 presents the results. We find that summaries from INFOSUMM-0.5B outperform PEGASUS$_{SFT}$ across all dimensions, and are considered to be more faithful than ChatGPT generated summaries. In Appendix E.2, we additionally conduct pairwise human evaluation of INFOSUMM against the baselines. We find that summaries from INFOSUMM are at least of equal quality with ChatGPT for more than 80% of the time, and are preferred to PEGASUS$_{SFT}$ for more than 50% of the time, demonstrating the strong performance of our distilled model.

### 3.3 Generalizing to Unseen Domains

**Zero-shot Generalization**  Next, we evaluate models on their generalization capability to benchmarks in unseen domains, specifically for WikiHow (Koupaee & Wang, 2018) and Reddit TIFU (Kim et al., 2019). We compare INFOSUMM-0.5B against three zero-shot baseline summarizers: Summary Loop, PEGASUS$_{CNN/DM}$ fine-tuned on CNN/DM train set, and zero-shot prompted ChatGPT.

In Table 2, we find that INFOSUMM-0.5B outperforms strong models in this setup, generating similar quality summaries as prompting ChatGPT. Notably, the results imply that INFOSUMM generalizes better than training on human-authored datasets: INFOSUMM-0.5B, trained on our distilled summaries, performs better in unseen domains than the same base model PEGASUS, fine-tuned on human-written summaries of CNN/DM.

**Fine-tuning to Unseen Domains**  One strength of a compact expert model is that it can be fine-tuned, to better follow the specific style and domain of benchmark references. This motivates us to consider another use-case of INFOSUMM, where a model is first distilled into a performant summarizer using synthetic data, then is further supervised on human-authored references to better adapt to an unseen domain.

In Table 3, while initial fine-tuning on CNN/DM (PEGASUS$_{CNN/DM\text{-}SFT}$) degrades the final model performance on unseen domains, INFOSUMM-0.5B improves over vanilla PEGASUS after fine-tuning.[3] We attribute this to the relatively narrow distribution of summary styles in commonly used summarization datasets (Tejaswin et al., 2021), including CNN/DM. As we show in §3.5, our dataset exhibits substantially larger scale, more extensive coverage of the summary space compared to the existing benchmarks, allowing the model to readily adapt to a specific summary style through fine-tuning.

### 3.4 Controllable Summarization

As the final application of INFOSUMM, we evaluate our model on controllable summarization. We use MACSum-Doc dataset (Zhang et al., 2023b), where summaries are collected from human annotators across 4 control dimensions: length (*short, medium, long*), extractiveness (*low, medium, high*), specificity (*medium, high*) and keywords. For baselines, we consider PEGASUS$_{MACSum}$ fine-tuned with the gold references in MACSum train set, along with zero-shot / few-shot prompted ChatGPT with 5 demonstrations sampled from MAC-Sum train set (for few-shot prompting, we use *gpt-3.5-turbo-16k*). To evaluate the controllability over length, extractiveness and specificity, we report control correlation (Zhang et al., 2023b), measuring how well a summary follows the given control code. In addition, we conduct human evaluation to assess the keyword usage and overall quality of generated summaries. See Appendix B and C for further evaluation details.

| Model | Control Correlation ↑ | | | Human Eval ↑ | |
|---|---|---|---|---|---|
| | *Len.* | *Ext.* | *Spe.* | *Keyword* | *Overall* |
| PEGASUS$_{MACSum}$ | 30.6 | **0.18** | 0.48 | 1.53 | 1.66 |
| ChatGPT | 29.8 | 0.01 | 0.07 | 2.65 | 2.58 |
| ChatGPT$_{5\text{-}shot}$ | 32.8 | 0.12 | 0.35 | **2.71** | **2.75** |
| INFOSUMM-0.5B | **37.6** | 0.16 | **1.82** | **2.71** | 2.70 |

Table 4: Results on controllable summarization. INFOSUMM-0.5B achieves better controllability across summary length, extractiveness, specificity than 5-shot prompted ChatGPT or human-supervised model.

INFOSUMM-0.5B significantly outperforms baselines in controllability across dimensions. While large-scale supervision is crucial for reliable controllability, human-authored train set is hard to scale. Accordingly, PEGASUS$_{MACSum}$ fine-tuned on the 4K samples of MACSum train set yields sub-optimal performance, although the references were curated by humans. Meanwhile, our model better correlates to the control codes than ChatGPT, even when the LLM was given 5-shot in-domain demonstrations. The result substantiates that while a textual description of constraints could signal some degree of control to LLMs, it may not suffice to control more sparse and fine-grained composition of attributes (Chen et al., 2023).

---

[3]We focus on reference-based evaluation, as fine-tuning does not improve G-Eval for both PEGASUS$_{CNN/DM}$ and INFOSUMM-0.5B.

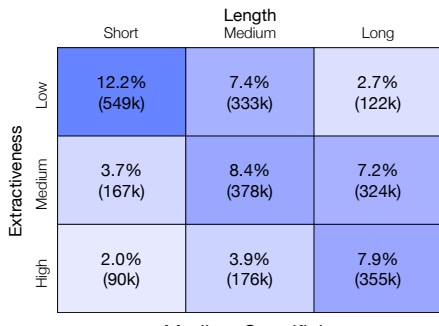
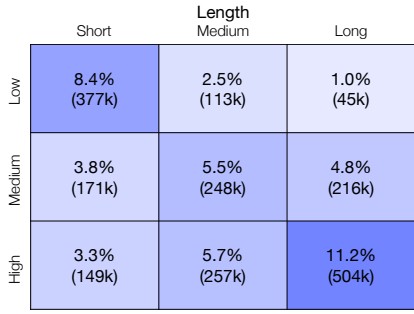

Figure 3: **Summary style distribution** of the distilled dataset from INFOSUMM. Compared to human-authored datasets (Appendix §E.4), our dataset entails substantially diverse coverage of summary styles, leading to a more robust and generalizable student.

## 3.5    Additional Analyses

**Re-ranking summaries**    The critic models defined in §2.1 play a pivotal role in INFOSUMM, identifying high-quality pairs for distillation. Thus, it is reasonable to ask whether the critic models are all we need, *i.e.,* strong performance can simply be achieved by re-ranking summaries generated from existing systems (*e.g.,* ChatGPT). To validate this hypothesis, we test *best-of-10* approach on top of ChatGPT and INFOSUMM-0.5B: we first sample 10 summaries per document, then output the generation with the largest sum of faithfulness and saliency score.

In human evaluation (Figure 2 and Appendix E.1), *best-of-10* on ChatGPT slightly improves its faithfulness and saliency, but undermines the fluency of the generated summaries. This is in contrast to our method, where *best-of-10* consistently improves in all evaluated aspects. The results show that (1) optimizing for PMI maximization indeed improves the human perception of faithfulness and saliency for both LLM and INFOSUMM generated summaries, but (2) unlike our model, a moderate sample size of 10 may not be enough for meaningful exploration with ChatGPT, as the model has not been trained to align with our search objective.

**Analyzing Data Diversity**    We directly evaluate the diversity of the distilled dataset against human-curated benchmarks widely used for news summarization. We first compare the summary length statistics and lexical diversity of each dataset (Table 5). To measure lexical diversity, we follow Jung et al. (2023) to report 2/3-gram entropy, along with mean segmented token type ratio (MSTTR; Torruella & Capsada (2013)) of the summaries. In addition, we

| Dataset | Length
Median (std) | Lexical Diversity
$H_2 \uparrow$ | $H_3 \uparrow$ | MSTTR $\uparrow$ |
|---|---|---|---|---|
| XSUM (0.2M) | 23.0 (5.7) | 16.9 | 19.9 | 55.5 |
| CNN/DM (0.3M) | 51.0 (22.9) | 18.4 | 22.0 | 54.3 |
| Gigaword (3.8M) | 8.0 (2.8) | 16.9 | 21.2 | 47.2 |
| Ours (4.5M) | 58.0 (**26.7**) | **18.8** | **23.2** | **57.7** |

Table 5: INFOSUMM yields a high-quality dataset with larger scale, more diverse summaries than existing benchmarks.

analyze the summary style distribution of each dataset (Figure 3 and Appendix E.4), by categorizing summaries into 18 style groups proposed in MACSum.

The results demonstrate that our dataset, as a purely synthetic corpus, is not only larger in sample size but also is substantially more diverse than existing datasets. As shown in Appendix E.4, human-authored datasets are typically skewed to a narrow region of style distribution—in XSUM, more than 70% of the reference summaries fall into short, less extractive and less specific group. Our dataset, on the other hand, covers significantly broader region of summary styles, along with consistently higher lexical diversity.

**Analyzing Sampling Efficiency**    In Appendix E.3, we conduct an ablation study on INFOSUMM, specifically focusing on the sampling efficiency of the framework (*i.e.,* the ratio of candidate summary pairs that pass all the critics). We leave the full results in Table 10, and summarize the results here. First, we find that the sampling efficiency of initial teacher prior to expert iteration is only 0.9%, indicating the importance of expert iteration for large-scale

distillation. However, we also find that multiple steps of expert iteration can over-optimize the teacher, leading to less diversity in generated data despite better sampling efficiency. We also find that PMI-maximization decoding (Eq. 5) improves the sampling efficiency by 4% compared to temperature-based sampling, representing its usefulness as inference-time algorithm to efficiently search for high-quality samples. See Appendix 3.5 for additional ablation results that focus on distilled model performance.

**Analyzing Effect of Expert Iteration** We also test whether expert iteration indeed improves the performance of the distilled summarizer. Specifically, we train an additional summarizer by directly fine-tuning PEGASUS-large on $D_{init}$, the initial dataset of 140k document-summary pairs generated from the off-the-shelf teacher $M_{init}$.

| Dataset | CNN/DM | | XSUM | |
|---|---|---|---|---|
| Model | *R-L* | *G-E* | *R-L* | *G-E* |
| No expert iteration | 35.8 | 4.07 | 82.5 | 4.05 |
| INFOSUMM-0.5B | **38.4** | **4.38** | **28.2** | **4.21** |

Table 6: Ablation results on expert iteration.

The performance of this configuration against the full INFOSUMM is shown in Table 6. INFOSUMM, trained with the large-scale summarization dataset generated by the improved teacher, yields consistently better scores in both reference-based (ROUGE-L) and reference-free evaluation (G-Eval). Surprisingly however, compared to the various unsupervised summarization models in Table 1, our model trained without expert iteration already outperforms majority of baselines in both CNN/DM and XSUM. The result shows that while expert iteration clearly benefits the student by scaling the training data, distilling with the information maximizing objective is as important to yield a reliable summarizer.

**Does PMI align with human evaluation?** We have shown through *best-of-n* analysis that optimizing for the PMI between the document and summary improves human evaluation results. In Appendix D, we further verify this by comparing the human-judged quality of references in XSUM against the PMI values estimated by the two critics of INFOSUMM. Overall, we find that the estimated PMI is an excellent predictor of human-evaluated quality, especially in the two tails of the score distribution (*i.e.,* when the pair is certainly of low-quality or high-quality). We also find that PMI estimation can often reveal annotation error in the widely-used dataset, indicating that our proposed objective can serve as a useful tool for filtering high-quality data for summarization.

# 4 Related Works

**Unsupervised Summarization** Prior approaches to unsupervised abstractive summarization have focused on devising a proxy task – *e.g.,* reconstruction of the original document – that may guide the model towards the desired behavior (Baziotis et al., 2019; Févry & Phang, 2018; Laban et al., 2020). While these methods typically require carefully designed training loop or reward shaping process (Yang et al., 2020), their performance often fall behind supervised models. Apart from the conventional methods, recent findings report that LLMs such as ChatGPT excel at summarization, surpassing the quality of human-supervised models without task-specific fine-tuning (Goyal et al., 2023; Zhang et al., 2023a). Subsequent works also show that a compact summarizer can be trained by imitating LLM-generated summaries (Sclar et al., 2022; Xu et al., 2023). Beyond LLM distillation, Jung et al. (2023) shares similar motivation to ours, presenting Impossible Distillation that distills a strong task model from a small, off-the-shelf teacher model. While Impossible Distillation is only applicable to sentence level tasks and requires a supervised NLI model, we target a more complex task of abstractive summarization with document-level inputs and operate entirely without human supervision.

**Generating Data with Language Models** Beyond summarization, a growing line of works proposes to directly generate a dataset using language models, tailored to specific domains and tasks such as commonsense knowledge (West et al., 2022; Brahman et al., 2023), mathematical / textual reasoning (Yu et al., 2023a; Mukherjee et al., 2023) and social dialogues (Kim et al., 2023). Nonetheless, challenges abound in automatic data generation – while the quality of data is a critical factor in downstream performance (Gunasekar et al.,

| | |
|---|---|
| **Document** | Police are searching for two missing teens believed to be together who disappeared after reportedly threatening to hurt themselves. Erika R—, 14, of Holiday and Caleb B—, 13, of Tampa are dating and it is suspected that she was picked up in a car on her way to Tampa with him, said police. B—, a white male, left his home near Dale Mabry Avenue and Lois Avenue on Saturday morning, and police are concerned due to his age as well as threats of him harming himself and his girlfriend, according to ABC Action News. Erika R— (left), 14, of Holiday and Caleb B— (right), 13, of Tampa are dating and it is suspected that she was picked up in a car on her way to Tampa with her boyfriend, said police. Authorities are searching for the teens who disappeared after reportedly threatening to hurt themselves . It is not known what he was wearing when he left his residence. R—, a white female, was last seen at her home around 11.30pm on Friday wearing a long-sleeve t-shirt, low-top gray Converse sneakers and possibly blue jeans. The teen girl, who also threatened to harm herself, took $200 from her mother's purse along with her school backpack before leaving her residence on Westchester Drive, according to WTSP. She has scars on both arms, on her upper thighs and under her armpits from self-mutilation, as well as a red mole on the left side of her forehead. The teen girl (R— pictured above), who also threatened to harm herself, took $200 from her mother's purse along with her school backpack before leaving her residence on Westchester Drive. She has scars on both arms, on her upper thighs and under her armpits from self-mutilation, as well as a red mole on the left side of her forehead . B— (above), a white male, left his home near Dale Mabry Avenue and Lois Avenue on Saturday morning, and police are concerned due to his age as well as threats of him harming himself and his girlfriend . R— is 5'6' tall, has auburn hair, light brown eyes and is 120lb. B— is 5'4' tall, has brown hair and is 130lb. Pasco Sheriff's Office spokeswoman Melanie Snow said R— had moved to Tampa from Holiday about three weeks ago, according to the Tampa Bay Times. She said: "We don't think that she is in danger, but she is only 14 years old and away from home on her own accord. Her mother wants her home." |
| **Summary** | Police are searching for two missing teenagers, Erika R—, 14, of Holiday and Caleb B—, 13, of Tampa, who disappeared after reportedly threatening to harm themselves. R— was last seen at her home around 11:30pm on Friday wearing a long-sleeve t-shirt, low-top gray Converse sneakers, and possibly blue jeans. B— left his home near Dale Mabry Avenue and Lois Avenue on Saturday morning, and police are concerned due to his age as well as threats of him harming himself and his girlfriend. |

Table 7: Unconstrained summary generated by INFOSUMM-0.5B, an entirely self-supervised summarizer with 568M parameters. We randomly sample a CNN/DM article and anonymize names of non-public figures in the table. More samples in Appendix F.

2023), even the strongest LMs suffer from unexpected mistakes (Jones et al., 2023; Jung et al., 2022) and lack of diversity in its generations (Yu et al., 2023b). Several techniques have been introduced to improve the data quality, such as verifier-guided sampling (Uesato et al., 2022; Lightman et al., 2023) and attributed prompting (Yu et al., 2023b; Yue et al., 2023; Eldan & Li, 2023), albeit in limited domains. Aligning with these works, we show that an explicit formalization of the target task can significantly boost the quality of generated dataset, and further demonstrate that prompting a larger, stronger LLM is not the only way to distill a performant summarization model.

## 5  Conclusion

We propose INFOSUMM, a novel method to distill a performant summarizer based on the information maximizing objective for summarization. We find that INFOSUMM, without either human annotated references or gigantic LLM, often outperforms the strongest baselines across benchmarks in news summarization, zero-shot / fine-tuning adaptation to unseen domains, and controllable summarization. In addition, we produce a large-scale summarization dataset as a byproduct of INFOSUMM, demonstrating the most extensive coverage of summary style and lexical diversity compared to existing benchmarks widely used in prior works. INFOSUMM demonstrates a way to distill a powerful summarizer based on how we formally characterize summarization, rather than how a teacher model behaves when instructed for summarization.

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

## Acknowledgements

This work was funded in part by the DARPA MCS program through NIWC Pacific (N66001-19-2-4031), ONR N00014-24-1-2207 and IARPA HIATUS via 2022-22072200003.

## A    Implementation Details

### A.1    Generation Stage

We start off from Pythia-2.8B as the initial teacher model. Following Jung et al. (2023), we use a simple prompt $P$ formatted as "*{City Name}, ({Media Name}) −*" in order to generate news-style summary and article. 23 media names and 984 city names were collected by the authors to automatically generate diverse $P$. We find that this way of prompt construction not only encourages the fluency of LM, but also significantly improves the diversity of generation compared to sampling without a prompt. While generating both the summary and article, we use top-p threshold of 0.9 with temperature = 1.0. We first generate summary by sampling 1-5 sentences conditioned on $P$, where the number of sentences are randomly chosen. Then, the article is generated by autoregressively conditioned on both $P$ and the summary. We do not fix the number of sentences in the article, and generate until the max number of tokens are reached (1024 for our experiments).

### A.2    Filtering Stage

We use T5-large, a masked language model with 770M parameters as the backbone for the faithfulness and saliency critics. To determine critic thresholds, we run a series of small-scale experiments to generate 1K samples from $M_{LM}$ and manually inspect the summary quality. Based on the results, we set $\tau_S = \log 14$, $\tau_F = \log 1.7$ and $\tau_B = 0.2$. Intuitively, $\tau_S = \log 14$ constrains that when the summary length is 10% of the original document, the likelihood of accurately inferring the masked tokens has to be 40% higher when an MLM is provided with the summary. We qualitatively find that while the high threshold leads to low sampling efficiency with the initial teacher, it improves the quality of the distilled dataset and hence leads to better performance of the end-stage student model.

### A.3    Training and Evaluation

After expert iteration, we train PEGASUS-large, an encoder-decoder pre-trained LM with 568M parameters on the 4.5M samples distilled from the improved teacher $M_T$. We fine-tune the model for 2 epochs with batch size 64 and max target tokens = 128. For all other hyperparameters, we use the same setting as chosen in the original paper (Zhang et al., 2020a).

For our main experiments, we report ROUGE, BERTScore along with G-Eval. G-Eval is a model-based metric computed by prompting an LLM (*e.g.,* ChatGPT, GPT-4) with chain-of-thought style prompt for text evaluation. Specifically, we use GPT-4 as the base LM for G-Eval, averaging 1-5 Likert scale scores averaged across 4 dimensions (coherence, consistency, fluency and saliency), following the setup of the original paper (Liu et al., 2023b). To instruct GPT-4, we use the prompt in the official implementation. Although G-Eval shows substantially higher correlation with human judgements of summary quality compared to conventional metrics, Liu et al. (2023b) also reports that GPT-4 is biased toward LLM generations, assigning higher scores to summaries from ChatGPT than the baselines (including human-authored summaries). Indeed, we find that ChatGPT consistently obtains the highest G-Eval score among all baselines throughout our experiments, even though it underperforms the baselines in reference-based metrics.

## B Controllable Summarization Details

### B.1 Distilling for Controllable Summarization

INFOSUMM can be extended to controllable summarization, by additionally annotating the generated summaries with the corresponding control attributes. We use 4 dimensions of control attributes proposed in Zhang et al. (2023b), *i.e.,* length, extractiveness, specificity and keywords. For length, extractiveness and specificity, we follow the original setup of MACSum to define the respective metric function $m_{attr}$ that maps each summary to its corresponding scalar value. Specifically, for length, $m_{len}$ is the number of tokens in the summary. For extractiveness, $m_{ext}$ is the average of ROUGE-2 precision and ROUGE-3 precision of the generated summary against the input document. For specificity, $m_{spe}$ is defined as $(0.1 \times vb + 0.2 \times tok + 0.3 \times nn + 0.4 \times cd)/|x|_s$, where *vb, tok, nn, cd* represent the number of verbs, tokens, nouns and numerical tokens, and $|x|_s$ denotes the number of sentences in the summary. For keywords, we extract 1 or 2 keywords from the summary identified by an off-the-shelf keyword extraction tool (Grootendorst, 2020).

Based on the computed values from each metric function, we annotate summaries according to their length (short, medium, long), extractiveness (low, medium, high) and specificity (medium, high), followed by keywords in each summary. For example, to annotate the length of summary $y$, we define

$$length(y) = \begin{cases} short, & \text{if} \quad m_{len}(y) < \tau_{len,1} \\ medium, & \text{if} \quad \tau_{len,1} \leq m_{len}(y) < \tau_{len,2} \\ long, & \text{otherwise} \end{cases} \tag{8}$$

Ideally, the thresholds $\tau_1$ and $\tau_2$ should reflect how humans perceive the control attributes – *e.g.,* how short a summary should be in order to be perceived as short by humans. To this end, we define the thresholds based on the statistics of human-written summaries in MACSum train set. For example, the threshold $\tau_{len,1}$ between the short and medium length summaries is defined as the median length of *short summaries* and *medium length summaries* authored by human annotators. The specific values of the thresholds are as shown in Table 8.

| Control attributes | $\tau_1$ | $\tau_2$ |
|---|---|---|
| Length | 38 | 69 |
| Extractiveness | 0.34 | 0.51 |
| Specificity | - | 4.8 |

Table 8: Threshold values defined for control attribute annotation.

We annotate 1M subset of distilled dataset following the above process. Then, we train the student model on the distilled dataset, to generate a controlled summary when it is prompted with a specific instruction for control attributes *(e.g. Generate a long summary with low extractiveness and high specificity, focusing on given keywords)*. We provide examples of controlled summaries generated by INFOSUMM-0.5B in Appendix F.

### B.2 Evaluation for Controllable Summarization

For quantifiable attributes *i.e.,* length, extractiveness and specificity, we report the control correlation (CC) of each summarization system. Control correlation measures how well a system follows the given instruction for a specific control dimension. Specifically, for a control attribute *(e.g.,* length) with a control value pair $[v_1, v_2]$ *(e.g.,* short, medium), we first generate two summaries $y_1$ and $y_2$ for the same document but with the different control values $v_1$ and $v_2$, while all other attributes are unchanged. Then, CC is defined as

$$CC_{attr}(y_1, y_2) = \frac{m_{attr}(y_1) - m_{attr}(y_2)}{d(v_1, v_2)} \tag{9}$$

where $d(v_1, v_2)$ defines the distance between the two control values, *e.g.,* $d(short, medium)$ = 1, $d(long, short) = -2$. Note that CC can be either positive or negative; when CC is negative, it indicates that the model has a negative correlation with the control instruction. We evaluate the system's average CC over a control dimension as the arithmetic mean over all samples in MACSum test set. For keyword and overall summary quality evaluation, we conduct human evaluation; see details in Appendix C.

# C Human Evaluation Details

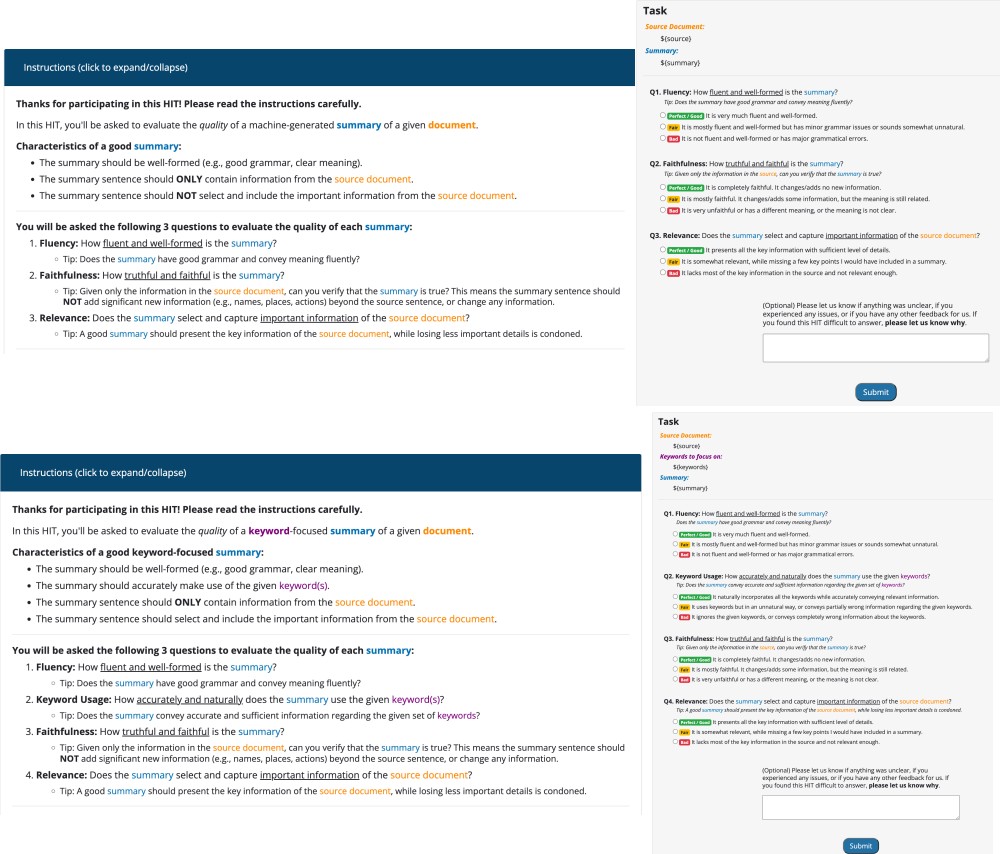

Figure 4: (Upper) Human evaluation template for unconstrained summary evaluation. (Lower) Human evaluation template for controllable summary evaluation.

For both unconstrained summary evaluation and controllable summary evaluation, human annotators are recruited from Amazon Mechanical Turk (MTurk) with an IRB approval. In unconstrained news summary evaluation, we generate summaries for 200 CNN/DM articles using each system, then ask 6 annotators to evaluate them across 3 evaluation dimensions (fluency, faithfulness and saliency). For controllable summary evaluation, we sample 200 documents (along with the control attributes) from MACSUM-Doc test set. We ask 6 annotators to evaluate the generated summaries for their keyword usage and overall quality (averaged across fluency, faithfulness and saliency). We compensate annotators with an hourly wage of $20.

# D Does PMI align with Human Judgements of Summary Quality?

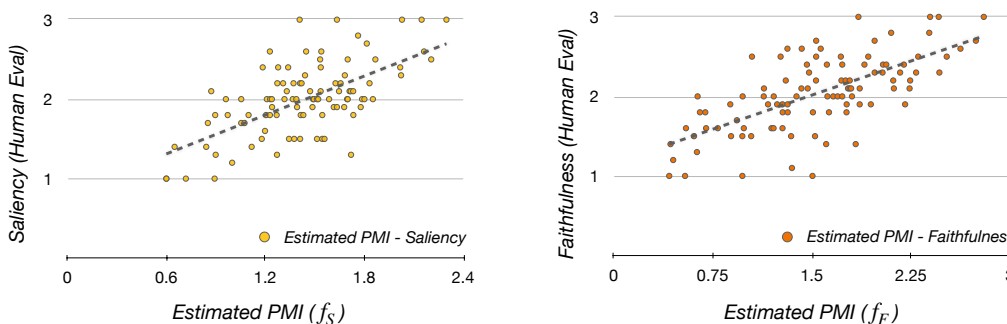

Figure 5: Human-evaluated quality of summary vs. estimated PMI by the saliency and faithfulness critics in INFOSUMM.

In this section, we further verify that PMI maximization under a length constraint aligns with the human-perceived quality of the summary. Following the same setup as in the main section, we conduct human evaluation to evaluate the quality of 100 reference summaries random-sampled from XSUM dataset. Then, we plot the human-judged saliency (faithfulness) of each summary against the PMI estimated by our proposed saliency (faithfulness) critic. We specifically choose XSUM because its reference summaries are bounded to be a single sentence, making it easier to control the confounding effect of length.

We present the results in Figure 5. In both dimensions, the estimated PMI exhibits positive correlation with the human-judged quality of reference summaries; maximizing PMI leads to more salient and faithful summaries, judged by humans. In addition, we find that particularly low value of estimated PMI often indicates annotation error – *e.g.,* the reference summary is completely irrelevant to the document, stating *"All images are copyrighted"*. This finding shows that even the widely-used summarization benchmarks are noisy, and PMI estimation can serve a useful tool for data cleaning prior to supervising a summarizer.

Notably, PMI becomes a better predictor of the human-judged quality in the two tails of the score distribution (*i.e.,* when the document-summary pair is certainly low-quality or high-quality), while the prediction gets slightly noisier in the middle range – this is expected, as even the human annotators show less agreement for the pairs with ambiguous quality. In fact, the result supports our choice of expert iteration as the learning algorithm - while directly optimizing for PMI with online learning may include training with the noisy reward in the middle, our method discards those pairs with susceptible estimated quality, training with only the high-quality samples we are more confident about.

# E  Additional Results

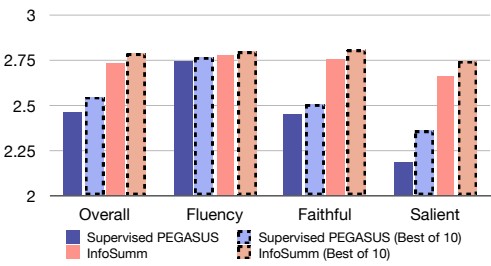

Figure 6: Additional human evaluation results with *best-of-10* on top of PEGASUS$_{SFT}$.

### E.1  Re-ranking summaries from the human-supervised model

In Figure 6, we provide human evaluation results of *best-of-10* summaries sampled from PEGASUS$_{SFT}$, a supervised model trained on CNN/DM train set, and compare it against INFOSUMM. While re-ranking the supervised model's summaries consistently improves the quality of summary across fluency, faithfulness and saliency, it still substantially falls behind INFOSUMM, indicating that the proposed information maximizing objective provides better supervision signal than imitating reference summaries of CNN/DM dataset.

### E.2  Pairwise human evaluation

To better compare the quality of summaries, we provide pairwise human evaluation results. Following the setup of prior works (Goyal et al., 2023; Stiennon et al., 2022), we ask 6 annotators to determine which summary is better. We use 200 CNN/DM articles and compare INFOSUMM-0.5B, ChatGPT and in-domain supervised PEGASUS$_{SFT}$, with and without re-ranking.

| No re-ranking: left wins / ties / loses (%) | |
| --- | --- |
| INFOSUMM vs. ChatGPT | INFOSUMM vs. PEGASUS$_{SFT}$ |
| 14.5 / 70.2 / 15.3 | 50.5 / 26.0 / 23.5 |
| **Best-of-10: left wins / ties / loses (%)** | |
| INFOSUMM vs. ChatGPT | INFOSUMM vs. PEGASUS$_{SFT}$ |
| 19.9 / 64.5 / 15.6 | 52.0 / 23.2 / 24.8 |

Table 9: Pairwise human evaluation results on CNN/DM.

The results are shown in Table 9. We find that summaries from INFOSUMM are evaluated to be at least equal quality with ChatGPT for more than 80% of the time, both with and without re-ranking. Our model is consistently preferred to PEGASUS$_{SFT}$ supervised on CNN/DM train set, winning for more than 50% of the samples in both settings.

### E.3  Sampling Efficiency Analysis

We conduct an ablation study on INFOSUMM, focusing on the sampling efficiency of the framework in Table 10. Concretely, we quantify the sampling efficiency as the ratio of candidate summary pairs generated by the teacher that pass all the critics.

| Model | Sampling Efficiency |
| --- | --- |
| No expert iteration | 0.9% |
| 2 expert iteration steps | 59.8% |
| No PMI-maximization decoding | 54.5% |
| INFOSUMM | 58.5% |

Table 10: Sampling efficiency analysis on INFOSUMM. Sampling efficiency of each configuration is defined as the ratio of the generated pairs that pass the critics.

First, we ablate the expert iteration and directly measure the sampling efficiency of the initial teacher $M_{init}$ (*No expert iteration*). As expected, the off-the-shelf LM's sampling efficiency is near zero, indicating the importance of expert iteration for large-scale distillation. However, we also find that multiple steps of expert iteration can lead to over-optimizing the data generator. While 2 steps of expert iteration yields slightly better efficiency than a single step (*2 expert iteration steps*),

we qualitatively find that it significantly reduces the diversity of generated data due to over-fitting the teacher model. We also consider an ablation (*No PMI-maximization decoding*) on our decoding algorithm by replacing it with Nucleus Sampling. The sampling efficiency in this case degrades by 4% than the original pipeline, attesting to the usefulness of the inference-time algorithm to efficiently search for high-quality samples.

### E.4 Baseline summary style distributions

In Figure 7, we plot the summary style distribution of 4 widely-used summarization datasets—CNN/DM, XSUM, Gigaword and WikiHow. Note that all these datasets were curated by humans and are of large-scale, consisting of least 200K train samples up to 3.8M samples in Gigaword. Nonetheless, compared to INFOSUMM, the reference summaries in each dataset are skewed to distinctive regions of summary style.

## F   Generation Samples

In Table 11-14, we provide unconstrained / controlled summaries generated by INFOSUMM-0.5B for non-cherry-picked XSUM, CNN/DM and WikiHow documents. To demonstrate controllability, we provide the control instruction (if applicable) along with the corresponding summary to the document.

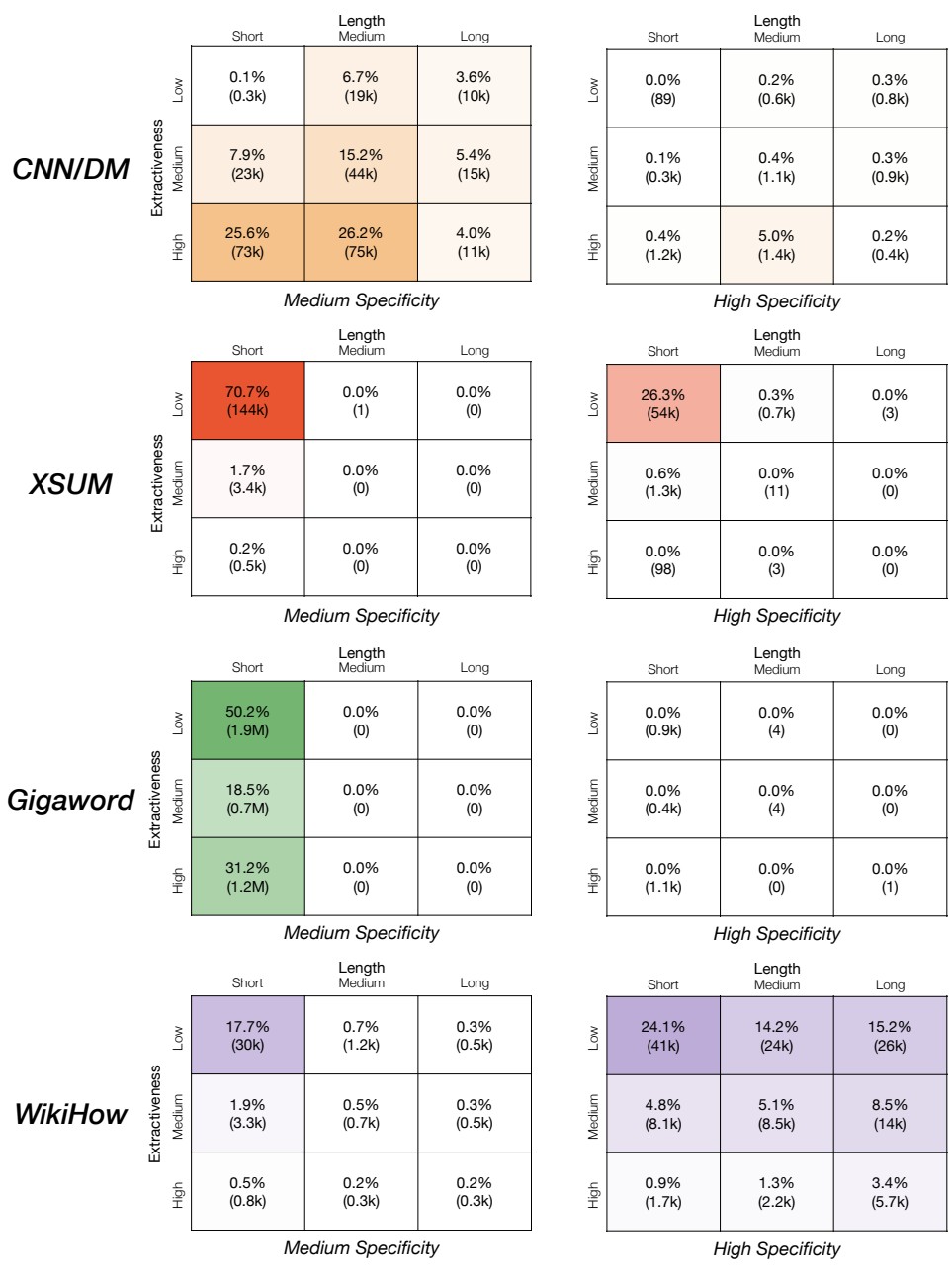

Figure 7: Summary style distribution of 4 commonly-used summarization datasets.

| | |
|---|---|
| **Document** | Police are searching for two missing teens believed to be together who disappeared after reportedly threatening to hurt themselves. Erika R—-, 14, of Holiday and Caleb B—-, 13, of Tampa are dating and it is suspected that she was picked up in a car on her way to Tampa with him, said police. B—-, a white male, left his home near Dale Mabry Avenue and Lois Avenue on Saturday morning, and police are concerned due to his age as well as threats of him harming himself and his girlfriend, according to ABC Action News. Erika R—- (left), 14, of Holiday and Caleb B—- (right), 13, of Tampa are dating and it is suspected that she was picked up in a car on her way to Tampa with her boyfriend, said police. Authorities are searching for the teens who disappeared after reportedly threatening to hurt themselves . It is not known what he was wearing when he left his residence. R—-, a white female, was last seen at her home around 11.30pm on Friday wearing a long-sleeve t-shirt, low-top gray Converse sneakers and possibly blue jeans. The teen girl, who also threatened to harm herself, took $200 from her mother's purse along with her school backpack before leaving her residence on Westchester Drive, according to WTSP. She has scars on both arms, on her upper thighs and under her armpits from self-mutilation, as well as a red mole on the left side of her forehead. The teen girl (R—- pictured above), who also threatened to harm herself, took $200 from her mother's purse along with her school backpack before leaving her residence on Westchester Drive. She has scars on both arms, on her upper thighs and under her armpits from self-mutilation, as well as a red mole on the left side of her forehead . B—- (above), a white male, left his home near Dale Mabry Avenue and Lois Avenue on Saturday morning, and police are concerned due to his age as well as threats of him harming himself and his girlfriend . R—- is 5'6' tall, has auburn hair, light brown eyes and is 120lb. B—- is 5'4' tall, has brown hair and is 130lb. Pasco Sheriff's Office spokeswoman Melanie Snow said R—- had moved to Tampa from Holiday about three weeks ago, according to the Tampa Bay Times. She said: "We don't think that she is in danger, but she is only 14 years old and away from home on her own accord. Her mother wants her home." |
| **Summary 1** | **Control Codes:** *Specificity - High / Length - Long*
**Output:** Police are searching for two missing teenagers, Erika R—-, 14, of Holiday and Caleb B—-, 13, of Tampa, who disappeared after reportedly threatening to harm themselves. R—- was last seen at her home around 11:30pm on Friday wearing a long-sleeve t-shirt, low-top gray Converse sneakers, and possibly blue jeans. B—- left his home near Dale Mabry Avenue and Lois Avenue on Saturday morning, and police are concerned due to his age and threats of harming himself and his girlfriend. R—- has scars on both arms, on her upper thighs, and under her armpits from self-mutilation, as well as a red mole on the left side of her forehead. |
| **Summary 2** | **Control Codes:** *Keywords - 200 / Length - Short*
**Output:** Police are searching for two missing teenagers, Erika R—-, 14, and Caleb B—-, 13, who disappeared after reportedly threatening to harm themselves, with R—- taking $200 from her mother's purse and B—- leaving his home near Dale Mabry Avenue and Lois Avenue on Saturday morning. |
| **Summary 3** | **Control Codes:** *Keywords - Melanie Snow / Length - Medium*
**Output:** Police in Tampa are searching for two missing teenagers, Erika R—-, 14, and Caleb B—-, 13, who are believed to be together. R—- was last seen at her home around 11:30pm on Friday wearing a long-sleeve t-shirt, low-top gray Converse sneakers, and possibly blue jeans. According to Pasco Sheriff's Office spokeswoman Melanie Snow, R—- had moved to Tampa from Holiday about three weeks ago and her mother wants her home. |

Table 11: Controlled summaries generated by INFOSUMM-0.5B for the same CNN/DM article as in Table 7. Names of non-public figures have been redacted in the table; the non-anonymized document can be found in the original dataset.

| | |
|---|---|
| **Document** | Media playback is not supported on this device

Farrell, 25, is set to move past 500 international points this weekend against Fiji, and is second in the England all-time list behind Wilkinson.

Asked whether Farrell could one day beat his record of 1,179 points, Wilkinson said: "I have no doubt".

"I would be very surprised if Owen Farrell didn't go on to score way, way more," he told BBC Sport. "500 points for a guy who is 25 years old, you don't have to do the maths but if he plays until he is 35, he will be in a good place." Wilkinson has been involved in the England camp this year as a kicking and skills consultant, and says he can identify with the way Saracens fly-half Farrell approaches the game.

"Sometimes when we are discussing things, you hear something that you definitely correspond with," said Wilkinson, who on Thursday became one of 12 new inductees into the World Rugby Hall of Fame.

"[Working with England] is a really exciting opportunity," Wilkinson, who scored the winning drop goal in the 2003 World Cup final, continued.

"But there is no telling or teaching, it's kind of a sharing, and therefore there has to be room in all of us to keep growing. I am also there to learn."

Wilkinson has also compared the current midfield combination of Farrell, who has scored 497 points for England, and fly-half George Ford to his partnership with inside centres Will Greenwood or Mike Catt.

"Both those guys were phenomenally important to me in my career, in the way they supported me, got the best out of me, and helped me to uncover more about myself," Wilkinson explained.

"With Owen and George, they are both very, very open individuals, both very humble - but not because they have been taught what to say, but genuinely it's who they are. There is not a script being followed here. They are following enthusiasm, passion and serious devotion."

"It's very similar when you mention names like Mike Catt and Will Greenwood, it all makes perfect sense. It's just about having good people in there."

You can hear more from Jonny Wilkinson on the Matt Dawson Rugby Show on BBC Radio 5 live from 19:30 GMT on Thursday, 17 November. |
| **Summary** | Owen Farrell, 25, is set to move past 500 international points this weekend against Fiji, and is second in the England all-time list behind Jonny Wilkinson. Wilkinson has been involved in the England camp this year as a kicking and skills consultant, and has identified with Farrell's approach to the game. Wilkinson has compared the current midfield combination of Farrell and fly-half George Ford to his partnership with inside centers Will Greenwood or Mike Catt, citing their enthusiasm, passion, and devotion. |

Table 12: Unconstrained summary generated by INFOSUMM-0.5B for a random-sampled XSUM article.

| | |
|---|---|
| **Document** | Media playback is not supported on this device
Farrell, 25, is set to move past 500 international points this weekend against Fiji, and is second in the England all-time list behind Wilkinson.
Asked whether Farrell could one day beat his record of 1,179 points, Wilkinson said: "I have no doubt".
"I would be very surprised if Owen Farrell didń́ go on to score way, way more," he told BBC Sport. "500 points for a guy who is 25 years old, you don't have to do the maths but if he plays until he is 35, he will be in a good place." Wilkinson has been involved in the England camp this year as a kicking and skills consultant, and says he can identify with the way Saracens fly-half Farrell approaches the game.
"Sometimes when we are discussing things, you hear something that you definitely correspond with," said Wilkinson, who on Thursday became one of 12 new inductees into the World Rugby Hall of Fame.
"[Working with England] is a really exciting opportunity," Wilkinson, who scored the winning drop goal in the 2003 World Cup final, continued.
"But there is no telling or teaching, it's kind of a sharing, and therefore there has to be room in all of us to keep growing. I am also there to learn."
Wilkinson has also compared the current midfield combination of Farrell, who has scored 497 points for England, and fly-half George Ford to his partnership with inside centres Will Greenwood or Mike Catt.
"Both those guys were phenomenally important to me in my career, in the way they supported me, got the best out of me, and helped me to uncover more about myself," Wilkinson explained.
"With Owen and George, they are both very, very open individuals, both very humble - but not because they have been taught what to say, but genuinely itś who they are. There is not a script being followed here. They are following enthusiasm, passion and serious devotion."
"It's very similar when you mention names like Mike Catt and Will Greenwood, it all makes perfect sense. It's just about having good people in there."
You can hear more from Jonny Wilkinson on the Matt Dawson Rugby Show on BBC Radio 5 live from 19:30 GMT on Thursday, 17 November. |
| **Summary 1** | **Control Codes:** *Extractiveness - High / Length - Short*
**Output:** Owen Farrell is set to move past 500 international points this weekend against Fiji, and is second in the England all-time list behind Jonny Wilkinson. Wilkinson has been involved in the England camp this year as a kicking and skills consultant, and has identified with Farrell's approach to the game. |
| **Summary 2** | **Control Codes:** *Keywords - 2003 World Cup / Length - Medium / Specificity - Medium*
**Output:** In 2003, Jonny Wilkinson scored the winning drop goal in the World Cup final for England. Wilkinson has been involved in the England camp this year as a kicking and skills consultant, and has identified with the way Saracens fly-half Owen Farrell approaches the game. Wilkinson has compared the current midfield combination of Farrell and fly-half George Ford to his partnership with inside centers Will Greenwood or Mike Catt. |
| **Summary 3** | **Control Codes:** *Keywords - 2003 World Cup / Length - Medium / Specificity - High*
**Output:** Owen Farrell is set to move past 500 international points this weekend against Fiji, and is second in the England all-time list behind Jonny Wilkinson. Wilkinson, who scored the winning drop goal in the 2003 World Cup final, has been involved in the England camp this year as a kicking and skills consultant and has identified with Farrell's approach to the game. Wilkinson has compared the current midfield combination of Farrell and fly-half George Ford to his partnership with inside centers Will Greenwood or Mike Catt. |

Table 13: Controlled summaries generated by INFOSUMM-0.5B for the same XSUM article as in Table 12.

| | |
|---|---|
| **Document** | Be sure to include coffee mugs or any other dirty cup that needs a good washing. Fill your kitchen sink with very hot water, enough water, that is more than half, but less than to the very top. Add a squirt of dish soap in the water, and on a sponge, and make sure your sponge is wet, with dish soap on it. Put in all the cups into the hot water in sink and start cleaning. Once you are done with the cups, you can wash your plates, cutlery and cooking utensils. Coffee stains, and other stains, need some work to get clean, so scrub a little harder, but also let all the cups soak in the hot water longer, for that gets the stains looser, and easier to get clean. Rinse off the soap and water, and make sure you rinse well in very hot water. Some people like to buy a rack to hang your cups, so the water drips off the cups. But, if you want your dishes to be done quickly, hang your cups on a rack. When you're done washing all the dishes, wipe the dishes with a cloth towel. You might want to wash your towel after drying the washer. Put away your clean cups, and other dishes in a cupboard, or somewhere safe from getting dirty, and where you can grab one with out breaking a cup or other dish. Purchase and place a kitchen rug near the sink. If you drop a cup during the washing, it's less likely to be broken and one under where you will gather cups from time to time. |
| **Summary 1** | **Control Codes:** *None (Unconstrained generation)*
**Output:** To clean coffee mugs and other dirty cups, fill a kitchen sink with hot water and add a squirt of dish soap. Scrub the cups with the soap and water, and then rinse them thoroughly with hot water. |
| **Summary 2** | **Control Codes:** *Keywords - rack / Extractiveness - Low / Length - Medium*
**Output:** To clean coffee mugs and other dirty cups, fill a sink with hot water and add a squirt of dish soap. Scrub the cups with a sponge and let them soak in the water for a few minutes, then rinse them off with hot water and a cloth towel. If you want your dishes to be done quickly, hang them on a rack, but if you want them to be clean quickly, wipe them with a cloth towel. |
| **Summary 3** | **Control Codes:** *Length - Long / Specificity - High*
**Output:** To clean coffee mugs and other dirty cups, fill a kitchen sink with hot water and add a squirt of dish soap. Scrub the cups with a sponge and let them soak in the hot water for a few minutes. Once the cups are clean, wash other dishes such as plates, cutlery, and cooking utensils. It's important to scrub the cups harder to remove coffee stains, but also let them soak in the hot water longer to loosen them. Rinse off the soap and water, and then wipe the dishes with a cloth towel; if you want your dishes to be done quickly, hang them on a rack, but if you want them to be done quickly, wipe them with a cloth towel. Finally, put away the clean cups and other dishes in a cupboard or somewhere safe from getting dirty, and place a kitchen rug near the sink to prevent spills. |

Table 14: Unconstrained / Controlled summaries generated by INFOSUMM-0.5B for a random-sampled WikiHow post.

# G    Limitations and Future Works

In this work, we primarily use expert iteration and distillation to optimize for the information-maximizing objective for summarization. While our approach allows the generation of a reusable, high-quality dataset for summarization, in principle, the proposed objective can be optimized using alternative training methods such as online reinforcement learning. Therefore, a straightforward extension to INFOSUMM would be to compare the performance and robustness of summarizers optimized through different learning techniques.

As conventional metrics such as ROUGE fail to accurately evaluate model outputs, recent works propose to further fine-tune summarizers with human preference data, *e.g.,* through reinforcement learning (Stiennon et al., 2022; Wu et al., 2021). Although we show through extensive experiments that INFOSUMM is capable of distilling a powerful summarizer from an off-the-shelf teacher, our search objective may fall short of representing the subtle nature of human preferences. Nonetheless, as demonstrated by the superior fine-tuning performance of INFOSUMM-0.5B to specific benchmarks, we envision that INFOSUMM can still function as a useful base model for learning from human feedback. In this scenario, INFOSUMM-0.5B can be harnessed as a strong base model for summarization, equipped with better initial performance and transferability than the models naively fine-tuned on human-authored references.

In addition, the high level methodology of INFOSUMM can be generalized to tasks beyond summarization. While different tasks would require different set of critic models, the method can be adopted to tasks where the correctness of input-output can be measured and evaluated, either through an external verifier (*e.g.,* commonsense reasoning; Liu et al., 2023a) or through symbolic execution (*e.g.,* code generation; Haluptzok et al., 2023). The distillation stage of INFOSUMM can also be improved by incorporating advanced learning techniques such as BRIO (Liu et al., 2022), beyond maximizing conditional likelihood of the summaries in the generated dataset.

