# OpenReview forum: "Information-Theoretic Distillation for Reference-less Summarization"
_colmweb.org/COLM/2024/Conference — COLM_

### Official Review · Reviewer_SZDS · 2024-05-09

**Rating:** 9
**Confidence:** 4
**Ethics Flag:** 1

**Summary:**

# A summary in bullet points
- The paper presents a method to distill a small state-of-the-art (unsupervised) summarizer (Pegasus 0.5B) from a small-ish LLM (Pythia 2.8B).
- Distillation involves generating a large-scale synthetic **English news summarization dataset** of 4.5 million samples.
- Synthetic data is generated by prompting a base LLM to continue from prefixes like "New York, (CNN)" to promote news-style generation.
- The method requires an off-the-shelf base LLM + a masked language model (T5-Large) but no human references.
- To improve the quality of the synthetic data the LLM is iteratively improved by combining 3 proposed "critics" and rejection sampling.
- The 3 critics model *saliency, faithfulness, and brevity* and are implemented via mutual information (probabilities are estimated by the pre-trained T5-Large model)
- The improved LLM is then used to generate a large-scale synthetic summarization dataset of 4.5 million article-summary pairs.
- 1 million of these pairs are posthoc annotated for controllable summarization.
- The paper evaluates the performance of training small summarizers (0.5B) on the synthetic dataset.
- Consistent improvements are shown across automatic benchmarks and human evaluation (in/out-domain and zero-shot.)
- Presented experiments show that the proposed summarizers surpass prior unsupervised state-of-the-art systems and are comparable
to summaries generated by ChatGPT (x300 larger).

**Questions To Authors:**

Great work! Most questions are implicit in the bullet points in "reasons to reject".

**Reasons To Accept:**

The paper contributes several impactful findings that are immediately valuable to the research community.
- An effective method to create large-scale quality synthetic summarization datasets without relying on existing labels or instructions.
- A novel summarization data generation model allowing further exploration.
- The first large-scale synthetic summarization dataset with promising results.
- Findings suggesting that synthetic summarization data generalizes better than existing summarization datasets with human-written summaries.
- State-of-the-art English summarizers for both generic and controllable summarization.

**Reasons To Reject:**

The paper lacks a few central details related to the contributions, methods, and experimentation.

- It is unclear if the LLM generator, dataset, and summarizers will be shared or whether the method is the contribution. The former would greatly strengthen the weight of this paper.
- There is a possibility of data contamination wrt. the main results on the CNN/DM and XSum datasets. The off-the-shelf LLM utilized to generate synthetic data has likely been trained on CNN, DailyMail, and BBC articles. If so this would mean that the generator LLM has been trained on the test sets and in turn, potentially generated synthetic dataset with the test set in it. This suspicion could be resolved by reporting the overlap between the synthetic data and the test sets but is not reported in the paper. My suspicion comes from the very high ROUGE scores on par with early SOTA **supervised** transformers models (BERTSumABS, Liu and Lapata, 2019).
- The process of generating and training on the synthetic dataset is unclear making it hard to reproduce.
  - Rejection sampling is central to the final size of the synthetic dataset (and the performance of the summarizers?), however, details are included nor cited. This may be simple but more clarify in this regard would significantly strengthen the paper.
  - The paper mentions that further "expert iterations" are bad, however, this claim is not quantified. How is the ideal number of iterations estimated?
  - The computational costs and requirements are unknown. At what point is this method more cost-efficient than simply running ChatGPT?
  - The target size of the synthetic dataset seems to be 10M but it's unclear why. The limitations and potential implications of the method are unclear.

---

> ### Author Rebuttal · Authors · 2024-05-27
>
> We sincerely appreciate the reviewer for both positive and thoughful comments! Below, we seek to address each of your concerns:
>
> ### Model & Dataset
> We agree with the reviewer that sharing both the models and dataset is an integral part of reproducible research. We will release the resources once they are free of anonymity constraints.
>
> ### Potential data contamination
> We follow the reviewer's suggestion to directly measure the token overlap between our generated dataset vs. CNN/DM and XSUM. Due to spatial constraints, please refer to the author response to Reviewer MKad for specific experimental setting and the results. Overall, the results show that the overlap of CNN/DM and XSUM against InfoSumm is nowhere larger than that of NewsCrawl, where there is no possibility of test set contamination.
>
> ### Rejection sampling details
> In fact, we included the implementation details in Appendix A, including the specifics of filtering stage for rejection sampling. During rejection sampling, we leave the samples that satisfy each threshold in all 3 critics, as specified in Eq (7). We will strengthen the reference from the main section to the Appendix in the revised version of the paper.
>
> ### Effect of multiple expert iteration steps
> We validate this effect in Appendix E.3, where we test with 2 rounds of expert iterations. While 2 expert iteration steps marginally improve the sampling efficiency, it significantly reduces the diversity of generations (e.g., MSTTR reduces from 57.7 to 52.9). This suggests that multiple rounds of expert iteration can over-optimize the teacher, compromising the quality of the dataset for better quantity (as measured by sampling efficiency).
>
> ### Computational Requirements
> We thank the reviewer for notifying this - as our framework only employs a relatively small model with <3B parameters, one would essentially require only 1 research-scale GPU with 11G VRAM. Compared to prompting LLMs, our framework is significantly cheaper in inference, while providing comparable performance to ChatGPT. In controllable summarization, one would roughly have to spend ~$3 per 1K inference for 5-shot prompting GPT-3.5-turbo (assuming each example consisting of 1024 token documents), only to see less controllability than InfoSumm, which only requires 568M model running on a local GPU.
>
> Lastly, we would like to refer the reviewer to Appendix G, where we discuss the limitation and future implications of InfoSumm. Again, we appreciate your thoughtful comments!

---

> > ### Comment · Reviewer_SZDS · 2024-06-05
> >
> > Thanks for addressing the comments and providing clarifications. I have not changed my score but my overall confidence this paper has been reinforced by your response. Below are a couple of questions which I'd be interested in.. Responses could be added to the paper but more likely suitable for a followup paper.
> >
> > - **Effect of multiple expert iteration steps**: I understand that further expert iterations had unwanted side effects. In the appendix you only report the sample efficiency of 0 and 2 steps. How quickly does it degrade at 3 or 4 steps? Also, what are these numbers 57.7 to 52.9? This is not reflected in Table 8. Quantifying this in greater detail would enable future research which could improve on this which in turn would strengthen the contribution this paper.
> > - **Computational costs**: It is clear that it's cheaper to run inference on a smaller LLM, but how would this compare to the costs of building the InfoSumm system? Conducting the expert iteration + generating the datasets over multiple rounds has non-trivial computational costs. It might not be straight forward to calculate costs, but reporting the flops or GPU hours would be insightful for replication and comparison in practical settings.
> > - **Dataset size**: The response does not elaborate on the size of the generated datasets. Could a larger dataset be generated? Is there a relationship between the filter-rate and the number of samples? Furthermore, is it possible to imagine generating datasets of this size to other domains? This would help promote future work in this (very interesting) research direction.
> >
> > Errors in the paper
> > - At second to last paragraph in section 3 Appendix E.4 is referenced. I think it was meant to be E.3? "algorithm to efficiently search for high-quality samples. See Appendix E.4 for additional ablation results that focus on distilled model performance.".

---

> > > ### Author Response · Authors · 2024-06-05
> > >
> > > Again we appreciate the thoughtful comments! The degradation of number (from 57.7 to 52.9) is measured through MSTTR (mean segmented token type ratio), representing lexical diversity of each dataset. We note that the degradation in MSTTR was not reported in the original version of the paper, and will include this in the revised version of the section.
> > >
> > > We agree that adopting our framework to other domains and tasks is an exciting and open research direction! While different domains may require different critics and objectives for expert iteration, some recent works on human alignment propose to self-train models without relying on human supervision (e.g. using self-evaluation capability of LLMs [1]). We are particularly excited to see if our framework extends to reasoning tasks where verification is critical.
> > >
> > > Regarding the dataset size (and accordingly the cost), the relationship between rejection sampling and the number of samples is quantified by the sampling efficiency throughout the paper. In our experiments, we generated 4.5M resulting samples with aggregated sampling efficiency of 58.5%. (We will specify the associated computational costs in the revised version of the paper.) While we did not scale up beyond this level as 4.5M samples sufficed to compete with the strong baselines, larger dataset can be generated by (1) simply running the pipeline more, or (2) increasing sampling efficiency, e.g. by lowering filtering rate. While in this work the filtering thresholds were considered as hyperparameters and were chosen through manual inspection of the outputs, an interesting future direction is to analyze the tradeoff between the precision and recall of synthetic data, and to explore the limit of synthetic data scale (e.g., what happens when we fine-tune on 100M pairs of summarization data?).
> > >
> > > Lastly, the reference to Appendix E.4 was intended — while in Appendix E.3 we evaluate sampling efficiency across expert iteration steps, in Appendix E.4 we compare the resulting model performance with and without expert iteration. We will clarity this in the revised version!
> > >
> > > [1] Yuan et al. Self-Rewarding Language Models. arXiv 2024

---

### Official Review · Reviewer_NLbV · 2024-05-11

**Rating:** 8
**Confidence:** 4
**Ethics Flag:** 1

**Summary:**

This paper focuses on information-theoretic distillation methods, investigating their application and effectiveness in machine learning models. The authors analyze the theoretical underpinnings of these distillation techniques, discuss their practical applications, and demonstrate their empirical performance through experiments. The goal is to contribute to the understanding of knowledge distillation in the context of reducing model size while maintaining accuracy.

**Reasons To Accept:**

1. The paper presents innovative approaches to information-theoretic distillation, advancing existing literature and proposing new strategies that could lead to more efficient models.
2. The authors provide a thorough theoretical analysis, supplementing it with empirical results that demonstrate the potential impact of the methods proposed.

**Reasons To Reject:**

The empirical evaluation could be expanded to include more diverse datasets or use cases to ensure the results' generalizability across different scenarios.

---

> ### Author Rebuttal · Authors · 2024-05-27
>
> We thank the reviewer for the positive and encouraging comments. We are happy that you found our method innovative and our analyses to be thorough both theoretically and empirically. Below, in addition to the 5 benchmarks evaluated in the main section of the paper, we additionally evaluate whether InfoSumm can generalize to dialogue summarization, a domain substantially different from news articles that InfoSumm was trained for.
>
> ---
> ### Application to another dataset
> We additionally report results on DialogSum [1] that aims to summarize real-life dialogue scenarios. Following the same setup with Section 3.3, we use ROUGE-L and G-Eval to compare InfoSumm against $\text{PEGASUS}_\text{CNN/DM}$ and $\text{gpt-3.5-turbo}$. The results are shown below:
>
> | **Datasets** | **R-L** | **G-E** |
> |--------------|:-------:|:-------:|
> | $\text{PEGASUS}_\text{CNN/DM}$ |   15.6  |   3.43  |
> | $\text{gpt-3.5-turbo}$ |  23.8  |  4.56  |
> | $\text{InfoSumm-0.5B}$ |  23.6 |  4.53  |
>
> Consistent to the results in unseen domain evaluation, we find that InfoSumm significantly outperforms human-supervised PEGASUS and is comparable to gpt-3.5-turbo, as measured by both reference-based / reference-less metrics.
>
> [1] Chen et al., DialogSum: A Real-Life Scenario Dialogue Summarization Dataset. Findings of ACL 2021

---

### Official Review · Reviewer_MKad · 2024-05-11

**Rating:** 8
**Confidence:** 4
**Ethics Flag:** 1

**Summary:**

This paper presents a method for distilling a summarizer based on the information-maximizing objective. The proposed method begins by self-training a teacher model from a T5-large, alongside self-supervised critics. Subsequently, the dataset is generated from this self-trained teacher model. Finally, the method fine-tunes a smaller student LM specialized for summarization. Experimental results demonstrate that the distilled summarizer exhibits superior performance compared to baseline models.

**Reasons To Accept:**

The paper is well-written and straightforward. The concept of distillation based on the information-maximizing objective for summarization is well-conceived. Furthermore, it provides an extensive and well-executed experimental validation of the proposed method, along with thorough analyses.

**Reasons To Reject:**

In this paper, Pythia-2.8B serves as the teacher model. The Pile dataset used for training Pythia-2.8B includes subsets like news and Wikipedia pages used in CNN/DM, XSum, and WikiHow. The lack of analysis related to these subsets raises concerns about the robustness of zero-shot and unseen domain experiments.

To underline the effectiveness of the proposed method, it is essential to expand the experimentation to include models such as Flan-T5, in addition to Pythia-2.8B. While Pythia-2.8B has shown promising results, the need to demonstrate the method's robustness across different models remains critical. This broader testing will provide a clearer indication of the method’s generalizability and effectiveness beyond its current scope.

The paper states that an initial dataset of 1.4 million was created for the experiment, but fails to clarify which seed document was used as input. This detail is crucial for understanding the dataset's composition and the experimental setup.

Moreover, the issue of generating candidate pairs and considering data diversity has not been addressed. The potential for duplication in the multiple datasets of document-summary pairs ($D_{summ}$) generated by the teacher model highlights the need for further analysis to ensure variety and prevent bias in the generated summaries.

---

> ### Author Rebuttal · Authors · 2024-05-27
>
> We thank the reviewer for both positive and thoughtful feedbacks! Below, we seek to address each of the reviewer's concerns.
> ### Benchmark exposure on the teacher model
> We agree that analyzing the possibility of contamination on existing summarization benchmarks is important for the robustness of reported result. To mitigate this concern, we following the suggestion of Reviewer SZDS to measure the token overlap between our generated dataset vs. CNN/DM and XSUM. We sample a subset of 100K summaries from InfoSumm, and compare it to CNN/DM and XSUM. For each of the sampled summaries, we measure the Max ROUGE-1 across all references in CNN/DM and XSUM. For comparison, we also report the ROUGE-1 between InfoSumm and a subset of NewsCrawl consisting of 5K articles between May~Aug 2023 (which is well after the knowledge cutoff of the Pile in 2020).
> |**Datasets**|**ROUGE**|
> |-|:-:|
> |CNN/DM|15.08 (± 2.25)|
> |XSUM|14.11 (± 2.13)|
> |NewsCrawl|14.65 (± 2.70)|
>
> The overlap of CNN/DM and XSUM against InfoSumm is nowhere larger than that of NewsCrawl, where there is no possibility of test set contamination; this suggests that superior performance of InfoSumm does not merely come from the memorization of the teacher model.
> ### Another base LM
> Following the reviewer’s comment, we add additional result using a different base LM than Pythia-2.8B. To make our setting further challenging than using Flan-T5 (which has been fine-tuned for instruction-following), we instead use off-the-shelf GPT2-XL with 1.5B parameters as the teacher model. Due to the spatial constraints, we leave the detailed experiment setup and the results in author response to Reviewer LVKS. Overall, we find that even when using a weaker teacher model such as GPT2, InfoSumm can generate high-quality pairs that pass all the defined critics.
> ### Source of input document
> As illustrated in Section 2.2, we do not make use of any input documents in both generating $D_{init}$ and $D_{summ}$. We remove dependence to an external source of human-written data by generating both the document and the corresponding summary from the base LM, using PMI maximization decoding.
> ### Diversity of generated data
> In fact, we conduct multiple analyses on the generated data diversity in Section 3.5 and Appendix E.5. We find that InfoSumm, as a purely generated corpus, is more diverse than human-curated datasets for summarization, both in terms of (1) lexical overlap between samples, and (2) summary style diversity.

---

> > ### Comment · Reviewer_MKad · 2024-06-05
> >
> > I want to thank the authors for their responses and clarifications. All questions raised by me are addressed. I have decided to increase my rating to 8.

---

> > > ### Author Response · Authors · 2024-06-05
> > >
> > > Thank you for reading through our comments and re-acknowledging our work!

---

### Official Review · Reviewer_LVKS · 2024-05-19

**Rating:** 8
**Confidence:** 4
**Ethics Flag:** 1

**Summary:**

The paper proposes a framework, InfoSumm, to distill a summarizer by leveraging the quantifiable information maximizing objective for long and document-level summarization (saliency, faithfulness and brevity), based on the dataset generated from an off-the-shelf LM. The proposed formulation outperforms the baseline models in both news summarization benchmarks and unseen domains, and generates controllable summaries across dimensions of length, extractiveness, specificity and keywords, all without relying on human-annotated references or large language models.

**Reasons To Accept:**

The paper conducted comprehensive and extensively experiments and analyses. The proposed framework demonstrates good performance to the in-domain reference-supervised models and shows better controllability than prompting ChatGPT. It also can be used for filtering high-quality data for summarization. One large-scale summarization dataset was generated as well in this work.

This paper is beneficial to readers, well-structured, and easy to read.

**Reasons To Reject:**

One might be curious why choosing Pythia-2.8B but not other LMs.

In section of Saliency on page 3, it is not fully convinced for saliency perspective by measuring how well an MLM can recover x from x_mask when given the summary y. The masked tokens might not represent important information.

Measuring brevity by the compression ration between the summary and the document is not fully representative.

Figure 1 was not mentioned in the main content.

On page 8 line 3, 'than 'should be replaced with 'then'.

---

> ### Author Rebuttal · Authors · 2024-05-27
>
> We sincerely appreciate the reviewer’s positive comments. Below, we address each of your questions:
>
> ---
> ### Choice of Base LM
> While we chose Pythia-2.8B as it is a small LM with sufficiently long position embedding for pair generation, our framework does not require using a specific base model. Here, we additionally report the sampling efficiency of our framework using GPT2-XL with 1.5B parameters as the teacher model. As the core contribution of InfoSumm lies in self-training the teacher model through the information-centric objective for summarization, we evaluate whether InfoSumm can self-train a summarization data generator even when using GPT2 as the base model, measured by sampling efficiency.
>
> |           | **Sampling Efficiency** |
> |-----------|:---:|
> | No expert iteration (GPT2-XL) | 0.5% |
> | **InfoSumm (GPT2-XL)** | 46.2% |
>
> As expected, zero-shot prompting GPT2-XL almost never generates a correct document-summary pair, as quantified by the sampling efficiency of 0.5%. After expert iteration, however, the sampling efficiency dramatically increases to 46.2%. The result shows that even when using a weaker teacher model such as GPT2, InfoSumm can generate high-quality pairs that pass all the defined critics.
>
> ---
> ### Masked tokens for Saliency & Compression Ratio for Brevity
> We agree with the reviewer that measuring mask likelihood given the summary may not suffice to fully capture the saliency, primarily when the mask selection is suboptimal. However, as shown in the denominator of Eq (1), the noise introduced by suboptimal masks are effectively mitigated by normalizing with the unconditional likelihood of the mask (i.e., without the summary). Despite its simplicity, we find that this normalization (along with a lightweight mask selection using TF-IDF) suffices to align with the human-perceived degree of saliency, as shown in Appendix D. For brevity, we would like to also note that while compression ratio is simple, it has been considered as an important attribute of summary style, and has been used as an scalable proxy of brevity in prior works (e.g., [1]).
>
> ---
> ### Figure 1, Typo
> Thank you for the pointer, we will update these in the revised version of the paper.
>
> [1] Grusky et al., Newsroom: A dataset of 1.3 Million Summaries with Diverse Extractive Strategies., NAACL 2018

---

### Comment · Area_Chair_c3S2 · 2024-06-03
**Author responses**

Hi reviewers,
Please take a look at author's responses and other reviews of the paper. If the rebuttals addressed your concerns, please let the authors know about this and update your review. If not, please continue to engage with the authors and the other reviewers in the discussion forum.

Overall reviewers are pretty positive about this paper. Concerns raised by multiple reviewers:
- Data contamination due to choice of base model or its training data (reviewers MKad, SZDS)
- Lack of experiments with more diverse datasets + models (reviewers NLbV, LVKS)

Thanks!

---

### Decision · Program_Chairs · 2024-07-10

**Decision:**

Accept

**Comment:**

The paper proposes infosumm, an information-theoretic distillation framework for summarization. The core idea is to train a strong teacher model using summarization task motivated information theoretic objectives. This teacher model is iteratively self-improved using expert iteration. Then, a smaller summarization model is distilled from this strong teacher. Crucially, this entire pipeline does not rely on any reference summaries.

Pro:
Generally, the reviewers agree that this is a strong paper. The proposed approach is interesting, performs well on the datasets evaluated, both in in-domain and out-of-domain scenarios. The experiments are exhaustive. The reviewers' issues about data contamination have been addressed in the rebuttal.

Cons:
The data processing pipeline to filter and create data should be shared as it seems non-trivial to reproduce.